

# Modulatory mechanisms of long-term volleyball practice on visuospatial working memory capacity: an fNIRS study

Wen Zhang[1,*], Jingru Liu[2,*], Fangfang Hu[1], Yang Liu[1] and Chao Kan[1]

[1] School of Physical Education, Shaanxi Normal University, Xi'an, Shaan'Xi, China
[2] Physical Education Department, Xi'an University of Posts and Telecommunications, Xi'an, Shaan'Xi, China
* These authors contributed equally to this work.

## ABSTRACT

**Objective:** Visuospatial working memory capacity is crucial for daily life and various cognitive processes. Previous studies have shown that physical training not only improves physical fitness but also visual visuospatial working memory capacity. However, few studies have explored visuospatial working memory improvement and brain plasticity changes with long-term volleyball exercise. Therefore, the purpose of this study was to gain insight into whether there is a relationship between long-term volleyball practice and visuospatial working memory and the effects on the prefrontal lobes of the brain.

**Methods:** Neural correlates of visuospatial working memory in elite ($n = 23$; raining age mean: $8.27 \pm 1.75$ years; age: $21.07 \pm 1.58$ years) and novice ($n = 23$; raining age mean: $1.81 \pm 0.56$ years; age: $20.53 \pm 1.36$ years) volleyball athletes are examined to uncover potential skill-based differences. Functional near-infrared spectroscopy (fNIRS) data from 46 participants performing visuospatial working memory test reveal compelling results.

**Results:** Compared with the novice group, the experts showed a higher accuracy rate (ACC) ($p = 0.021$) and shorter reaction time (RT) ($p = 0.019$) in the visuospatial working memory test. fNIRS data showed increased oxygen activation in the right dorsolateral prefrontal lobe ($p < 0.05$) and the right frontal region ($p < 0.05$).

**Conclusions:** Studies have shown that long-term volleyball training can significantly enhance individuals' visuospatial working memory capacity. This enhancement was mainly reflected in the fact that athletes who participated in long-term volleyball training demonstrated faster operational processing speed and higher accuracy in visuospatial working memory tasks, and plasticity changes in dorsolateral prefrontal and prefrontal pole regions. We also found no significant linear relationship between specific brain activation and behavioral performance in expert athletes.

## INTRODUCTION

Cognitive psychology, particularly in the domain of visuospatial memory, has received considerable attention due to its implications for understanding various cognitive

Corresponding authors
Yang Liu, liuyang0330@snnu.edu.cn
Chao Kan, kanchao@snnu.edu.cn

processes and disorders (*Glikmann-Johnston et al., 2019*; *Williams et al., 2006*). Visuospatial memory refers to the ability to process and recall spatial and visual information, which is crucial for everyday tasks such as navigation and object recognition. This cognitive function is closely linked to working memory, which serves as a temporary storage system for information that is actively manipulated or processed (*Wang, 2012*).

Visuospatial working memory (VSWM) is a crucial cognitive function that plays a significant role in sports, particularly in dynamic and fast-paced games such as volleyball. Volleyball requires athletes to process and respond to rapidly changing visual information, which is closely related to their visuospatial cognitive abilities. During the game, athletes must quickly search for relevant information on the court, including the position of the ball, its trajectory, and spatial-temporal data, such as the athletes' positions, from both short-term and long-term memory, in order to make accurate decisions. Studies have shown that elite volleyball players exhibit superior perceptual cognitive abilities, as reflected in their faster reaction times in visuospatial tasks compared to players with less experience (*Liu, 2019*). Therefore, analyzing the visuospatial working memory characteristics of volleyball players and verifying whether sports experience enhances visuospatial working memory capacity are central issues.

The brain plasticity hypothesis posits that exercise-induced cognitive improvements result from alterations in brain activation patterns (*Li et al., 2023*) through long-term specialized training, which not only enhances athletes' task performance but also induces changes in cortical processing patterns (*Wei & Li, 2018*). Specific practice enables expert athletes to develop specialized visuospatial working memory traits. It has been shown that expert football players invest fewer cognitive resources in specific ball stimuli (*e.g.*, passes and catches) and capture attention more quickly. The same holds for tennis players, who exhibit more efficient recall retrieval and more accurate recognition of game patterns (*Binbin, 2021*). All of these studies demonstrate that expert athletes are able to make rapid, top-down information-processing judgments and decisions, and that the transfer of domain-specific skills to general cognitive tasks is confirmed (*Voyer & Jansen, 2017*). The trend of activation changes in the task-related brain regions of athletes compared to novices was inconsistent, with diving (*Li, 2023*) and basketball (*Milton et al., 2007*) athletes exhibiting lower levels of activation during cognitive task processing. However, in golf (*Yu et al., 2022*), ice hockey (*Del Percio et al., 2019*), football (*Rennig et al., 2013*), and chess (*Alves et al., 2013*), some studies have shown higher cortical activation in experts compared to novices.

Visuospatial working memory (VSWM) is an important cognitive function that plays an important role in sports, especially in dynamic and fast-paced games such as volleyball. Volleyball requires athletes to process and respond to rapidly changing visual information, which is closely related to their visuospatial cognitive abilities. During the game, athletes need to quickly search for effective information on the court, such as the position of the ball, its trajectory, and spatial-temporal information such as the stance of the athletes on the court, (*Voyer & Jansen, 2017*; *Ansari et al., 2015*) to match and extract from short-term vision and long-term memory, so as to make the correct decision. Studies have shown that elite volleyball players exhibit greater perceptual cognitive abilities, which is reflected in

their superior reaction times in visuospatial tasks compared to those with less experience (*Liu, 2019*). Therefore, it is a central issue to analyse the visuospatial working memory characteristics of volleyball players and to verify whether sports experience facilitates the improvement of visuospatial working memory capacity.

However, other studies have concluded that there is no difference in visuospatial working memory capacity between expert athletes and novices. Athletes in chess, table tennis, and orienteering did not differ in behavioral data and EEG metrics between novices and experts, nor did they show a cognitive advantage (*Li & Smith, 2021*). *Alves et al. (2013)* focused on volleyball experts in non-specific, scenario-based tasks (executive control, visuospatial memory, and attention) (*Yang et al., 2020*). A study was conducted that found experts exhibit better behavioral performance, and that the brain may show similar activation patterns across different tasks. In summary, item type and motor experience contribute to the inconsistent results of neural mechanisms during athletes' performance of various tasks, and the mechanisms of cortical activation in expert volleyball players during visuospatial working memory tasks remain to be explored further.

Some studies have shown that, during cognitive processing, the prefrontal cortical region of the brain receives processed external information from other functional areas and subsequently integrates brain information, such as memories and intentions, to generate a rational plan (*Rennig et al., 2013*; *Yang et al., 2020*; *Courtney et al., 1996*). It has been found that the prefrontal lobe is involved in visuospatial working memory tasks, with the dorsolateral prefrontal cortex (DLPFC), the frontal pole area (FOA), and the ventral lateral prefrontal cortex (VLPFC) being associated with the retention and processing of working memory information, while the orbital frontal cortex (OFC) is involved in directional recognition (*Pinti et al., 2020*; *Guo et al., 2008*; *Zoudji, Thon & Debû, 2019*). Therefore, in the present study, the VLPFC, DLPFC, FOA, and OFC were selected to investigate the activation characteristics of brain oxygenated hemoglobin (Oxy-Hb) in volleyball players during the visuospatial working memory task (*Dunst et al., 2014*; *Hazarika & Dasgupta, 2018*; *Jost et al., 2011*), and to identify the locations of activated brain regions associated with the cognitive abilities involved in visuospatial working memory in expert volleyball athletes, in order to explore the mechanisms underlying brain region activation and provide a neurological basis for enhancing visuospatial working memory performance in athletes in the future.

# METHODS

## Participants

The experimental plan's sample size was estimated through use of G Power 3.1 software. An effect size of 0.90 is expected, with a set α of 0.05 and a statistical power of 1- β of 0.8. Demographic data for the participants is presented in Table 1. Referring to previous studies, we categorized experts and novices (*Swann, Moran & Piggott, 2015*), including 23 experts (age, 21.07 ± 1.58 years; mean years of training: 8.27 ± 1.75 years) and 23 novices (age, 20.53 ± 1.36 years; mean years of training: 1.81 ± 0.56 years). The experts are made up of athletes from the National Premier League, and the group of novices are all university students majoring in physical education in colleges and universities.
**Table 1 Basic information table of the subjects.**

|  | Expert group | Novice group |
|---|---|---|
| Number of people | 23 | 23 |
| Sports level | Master level & First-level | No sports level |
| Average age | 21.07 ± 1.58 | 20.53 ± 1.36 |
| Average years of training (years) | 8.27 ± 1.75 | 1.81 ± 0.56 |
| Weekly training frequency (hour/weeks) | 10.2 ± 1.5 | 1.2 ± 1.5 |

All novice athletes have never practiced volleyball training or any other formal sports training, nor have they undergone any lengthy (more than 1 month) or regular (on a weekly basis) training periods. The informed consent of all participants was obtained and documented (see Supplemental Materials). The informed consent of all participants was obtained and documented (see Supplemental Materials). This study complied with the Declaration of Helsinki and was approved by the ethics and morality committee of Shaanxi Normal University (Approval number: SNNU2023301).

The inclusion criteria for each group were: all participants had to be in good health, possess normal or corrected vision and not be color-blind or weak, have no mental illnesses, be right-handed, and receive payment for their participation at the end of the experiment. None of the participants reported having a history of neurological or psychiatric disorders. The participants were instructed to abstain from alcohol for 24 h and from caffeine-containing substances for 12 h before the experiment. The study followed the principles outlined in the Declaration of Helsinki.

## Visuospatial working memory testing

The Visuospatial Working Memory task is depicted in Fig. 1. The test program are written and presented in E-prime 3.0 software, and the visual spatial working memory task is a square composed of a matrix of 5 × 5 pieces, where the random six squares are white. A red "+" fixation point appears in the center of a white background (complete computer display) for 1,000 ms; then the memory stimulus image appears for 2,000 ms, then the 1,000 ms empty screen, and finally the target stimulus image (2,000 ms). Subjects were asked to judge whether the target stimulus picture was consistent with the memory stimulus picture, and press f consistently and j inconsistently. The visual spatial working memory task included 30 trains, with 20,000 ms for every five trains breaks. The response time and accuracy index required by the visual spatial working memory task were recorded, and the concentration change of oxyhemoglobin was recorded with near-infrared when completing the visual spatial working memory task, to reflect brain function and other indicators.

## fNIRS recordings

The fNIRS test: A continuous wave Shimadzu LABNIRS system (Shimadzu Corporation, Kyoto, Japan) was utilised to gather haemodynamic signals from specific brain areas whilst the subjects completed their tasks. In terms of channel layout, it includes eight light source
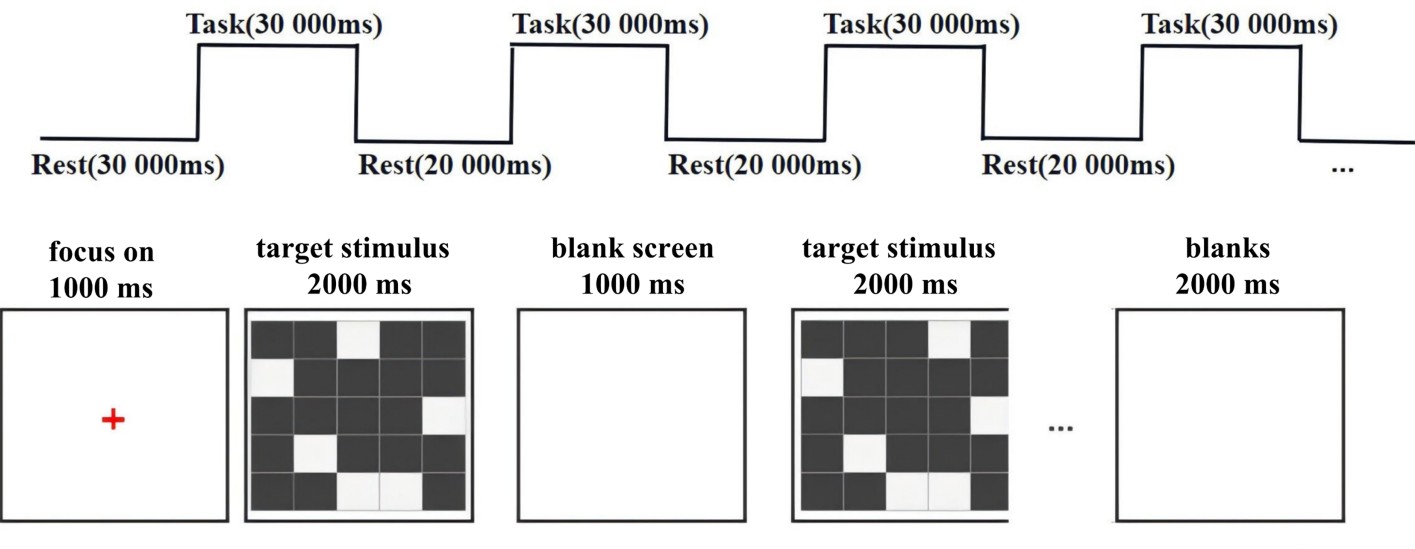

**Figure 1 Visuospatial working memory test procedure.** The visuospatial working memory task consisted of a 5 × 5 matrix consisting of a square in which six random squares were white. A red "+" gaze point would first appear in the center of a white background (complete with computer display) for 1,000 ms; this was followed by the presentation of a literacy stimulus picture for 2,000 ms, then a blank screen for 1,000 ms, and finally the target stimulus picture (presented for 2,000 ms). Subjects were asked to judge whether the target stimulus picture was congruent with the literacy stimulus picture, pressing the f key for congruence and the j key for incongruence. The visuospatial working memory task consisted of a total of 30 trails with a 20,000 ms rest every five trails.

probes and eight receiving probes (red is the light source emitter, blue is the light source receiver), which constitute 22 measurement channels with sampling frequency set to 13.3 Hz (see Fig. 2 and Table 2). From the existing anatomical calibration system (Anatomical Labeling Systems, LBPA40) To divide the areas of interest (region of interest, ROI), 8 ROI (Table 2): left dorsolateral prefrontal lobe (L-DLPFC); right dorsolateral prefrontal lobe (R-DLPFC); left ventral lateral prefrontal lobe (L-VLPFC); right ventral lateral prefrontal lobe (R-VLPFC); left frontopolar region (L-FOA); right frontal polar region (R-FOA); the left orbitofrontal region (L-OFC); right lateral orbitofrontal region (R-OFC); the above eight ROI were evenly distributed in the prefrontal lobe, Using the spatial registration of multichannel NIR data to MNI space. After the fNIRS recording, the probe position was determined using a 3D locator (FASTRAK; Polhemus, Colchester, VT, USA). The fNIRS channel position and the MNI spatial coordinates are registered by the probability registration method.

## Data collection and analysis

Behavioral data: The normal distribution of the measured data was tested using SPSS 26.0 and found to be greater than the 0.05 threshold, indicating that it followed a normal distribution. Independent sample t-tests were conducted on the subjects' correct rate and response time under the visuospatial working memory task to observe the differences in behavioral indicators at different levels and tasks.

fNIRS data: The raw data collected were resolved using the software supplied with the fNIRS device. Data processing was based on the NIRS_SPM toolbox of the MATLAB

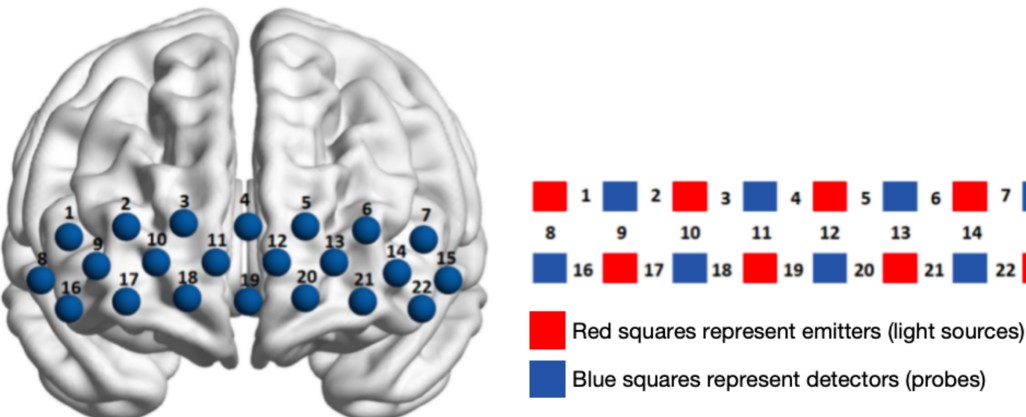

**Figure 2** Channel layout and calibration of brain regions in the prefrontal cortex.

**Table 2 Descriptive statistics behavioral measurements of spatial working memory tasks (M ± SD).**

| Brain regions | Correspondence channel |
|---|---|
| Right dorsolateral prefrontal lobe (R-DLPFC) | Ch1, Ch2, Ch9 |
| Left dorsolateral prefrontal lobe (L-DLPFC) | Ch6, Ch7, Ch14 |
| Right frontal pole area (R-FOA) | Ch3, Ch4, Ch10, Ch11 |
| Left frontal pole area (L-FOA) | Ch4, Ch5, Ch12, Ch13 |
| Right ventral lateral prefrontal (R-VLPFC) | Ch8, Ch16 |
| Left ventral lateral prefrontal lobe (L-VLPFC) | Ch15, Ch22 |
| Right orbitofrontal region (R-OFC) | Ch17, Ch18, Ch19 |
| Left orbitofrontal region (L-OFC) | Ch19, Ch20, Ch21 |

(R2013b) platform for analysis of fNIRS data (*Ye et al., 2008*). During NIRS experiments, there is a global drift due to the subject's head movement, vascular movement, blood pressure changes, physiological changes or instrument instability, resulting in a low-frequency bias. In addition, the amplitude of the global drift is usually comparable to the signal amplitude of the brain activation process. To eliminate the global trend and improve the signal-to-noise ratio, in this study, noise (head movement, heartbeat, breathing, *etc.*,) and drift were excluded according to the hemodynamic response function (HRF) and the wavelet-based minimum description length (Wavelet ___MDL) algorithm for de-trending (*Jang et al., 2009*), both of which have been demonstrated in related studies (*Brigadoi et al., 2014*). Specifically, the wavelet transform was used to decompose the NIR measurement data into global trends, haemodynamic signals and uncorrelated noise components at different scales. Meanwhile, the minimum description length (MDL) principle was used to avoid overfitting or underfitting of the global trend estimation. In addition, the pre-colouring method (*Friston et al., 1995*) was chosen for this study, where the kernel function can be either a Gaussian filter or a hemodynamic response function (HRF) low-pass filter. The HRF filter used in this paper is the preferred filter for NIR spectroscopic data because the transfer function of the HRF is based on the frequency of

neural signal modelling (*Schroeter et al., 2004*), which removes random noise generated by the instrument as well as physiological noise caused by heart rate, respiration, *etc.* For the construction of the design matrix based on the general linear model (GLM) (*McCormick et al., 1992*), the change in Hb O in each channel was calculated by the NIRS_SPM software and expressed as a beta value, and the beta value under the task conditions was evaluated as an indicator of the activation of the corresponding channel.

## RESULTS

### Behavioral results

Analysis of behavioural results under the visuo-spatial working memory task in volleyball players of different levels using independent samples t-test: accuracy in the novice group was significantly lower than in the expert group t (44) = 2.336, $p = 0.024$, Cohen's d = 0.69); response time was significantly greater than in the expert group t (44) = −2.149, ($p = 0.037$, Cohen's d = −0.63). The results showed a clear effect of expertise, with shorter RT and higher accuracy for experts than novices.

### fNIRS results

To investigate the cognitive and neural features of visual spatial working memory among volleyball players of varying abilities, we utilized the β values of each channel as the dependent variable and conducted an independent-samples t-test (Table 3).

As illustrated in Fig. 3. and Table 4. The results showed that the intensity of the visual spatial working memory task was in Ch1 (44): t = 3.218, $p = 0.002 < 0.05$, Cohen's d = 0.95; Ch2 (44): t = 3.261, $p = 0.002 < 0.05$, Cohen's d = 0.96; Ch9 (44): t = 3.059, $p = 0.004 < 0.05$, Cohen's d = 0.90, And Ch11 (44): t = 3.166, $p = 0.004 < 0.05$, Cohen's d = 0.93, Higher than in the novice group, according to the brain area corresponding to the channel, further reflecting the higher activation in the right dorsolateral prefrontal region and the right frontal polar region of the expert group than in the novice group.

### Correlations of fNIRS results and accuracy

To further test the correlation between behavioral indicators and fNIRS indicators, the behavioral performance of spatial memory and the brain region where the channel is located are correlated. Pearson's correlation analyzed the beta values of the four channels (Ch 1, Ch 2, Ch9 and Ch 11) at different levels, and the results are shown in Figs. 4–11.

As shown in Figs. 4–7, the expert group's accuracy rates were positively correlated with Ch1 (r = 0.38), Ch2 (r = 0.25, Ch9 (r = 0.37), and Ch11 (r = 0.34), but none of these correlations were found to be statistically significant ($p > 0.05$). On the other hand, the novice group's accuracy rates were positively correlated with Ch1 (r = 0.19) and negatively correlated with Ch2 (r = −0.03), Ch9 (r = −0.03), and Ch11 (r = −0.15), but none of these correlations were found to be statistically significant ($p > 0.05$) (as shown in Figs. 8–11)

## DISCUSSION

The aim of this exploratory study was to examine the potential relationship between long-term volleyball training and visuospatial working memory capacity. Functional

**Table 3 Descriptive statistical behavioural measures of spatial working memory tasks.** The reflection time and correctness of expert and novice groups in visuo-spatial tasks is described.

|  | Expert group | Novice group |
|---|---|---|
| Correct rate % | 0.83 ± 0.04* | 0.79 ± 0.06 |
| Reaction time/ms | 1,199.89 ± 162.25* | 1,289.91 ± 118.47 |

Note:
* $0.01 < p < 0.05$.

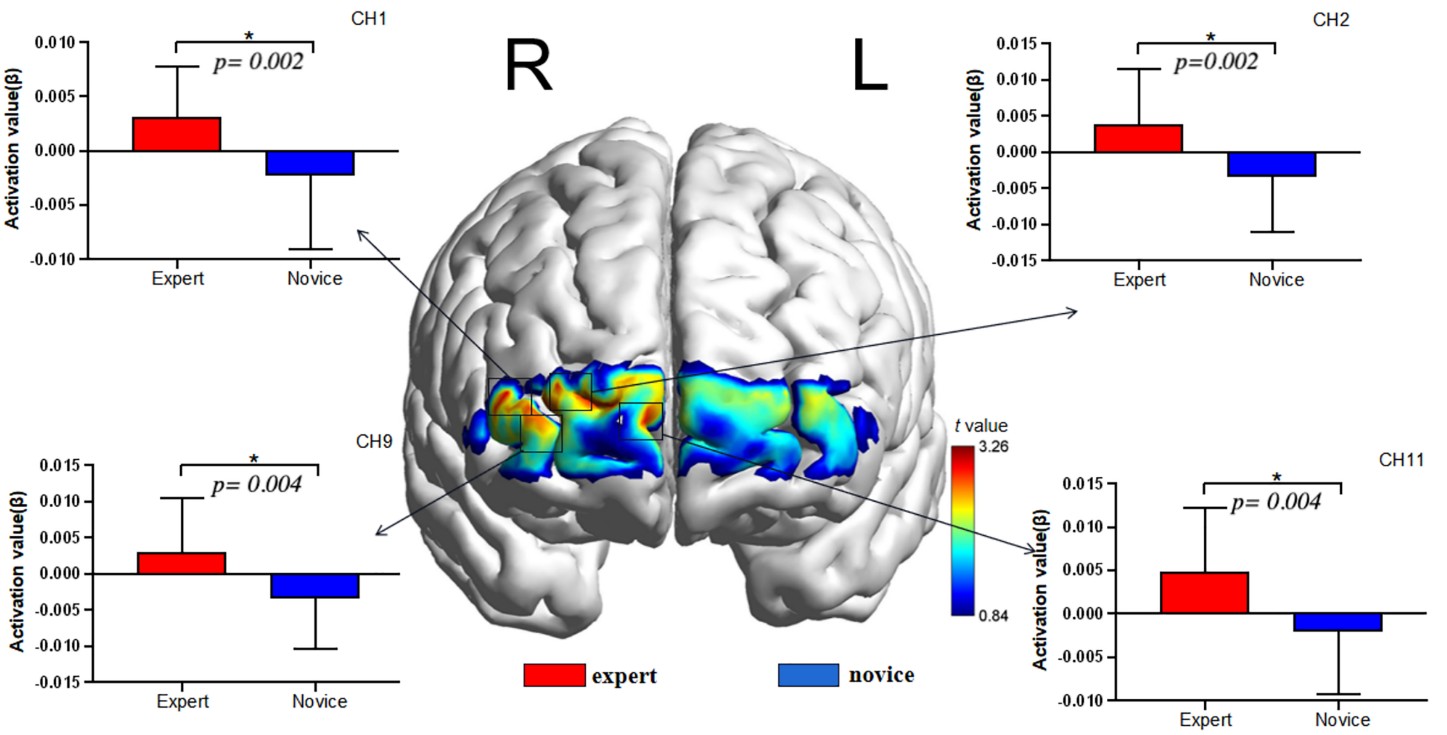

**Figure 3  fNIRS channel layout and distribution map.**               

**Table 4 Channel activation related to visuospatial memory test in volleyball players (mean ± standard deviation).**

| Channel | Expert ($n = 23$) | Novice ($n = 23$) | $t$ | $P$ |
|---|---|---|---|---|
| CH1 | 0.003156 ± 0.004616 | −0.002328 ± 0.006743 | 3.218 | 0.002 |
| CH2 | 0.003905 ± 0.007581 | −0.003407 ± 0.007624 | 3.261 | 0.002 |
| CH9 | 0.003030 ± 0.007385 | −0.003417 ± 0.006901 | 3.059 | 0.004 |
| CH11 | 0.004877 ± 0.007402 | −0.001998 ± 0.007326 | 3.166 | 0.004 |

near-infrared spectroscopy (fNIRS) was employed to incorporate a visuospatial working memory test as the task, allowing us to explore both behavioral and neural activation differences between expert and novice athletes. The findings suggest that long-term volleyball training enhances visuospatial working memory. Specifically, experts demonstrated shorter reaction times, higher accuracy, and increased activation in the
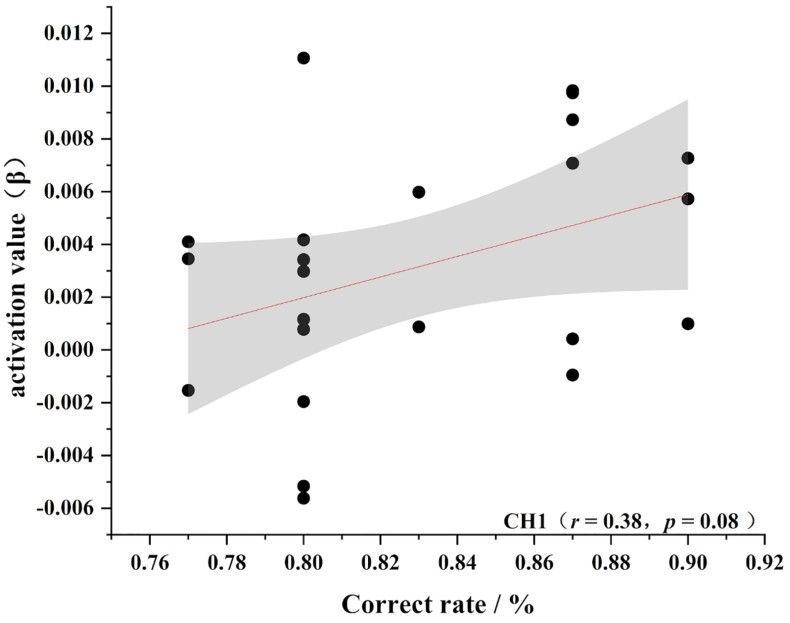

**Figure 4 Accuracy and relevance of experts group channel 1.** CH1 for channel 1.

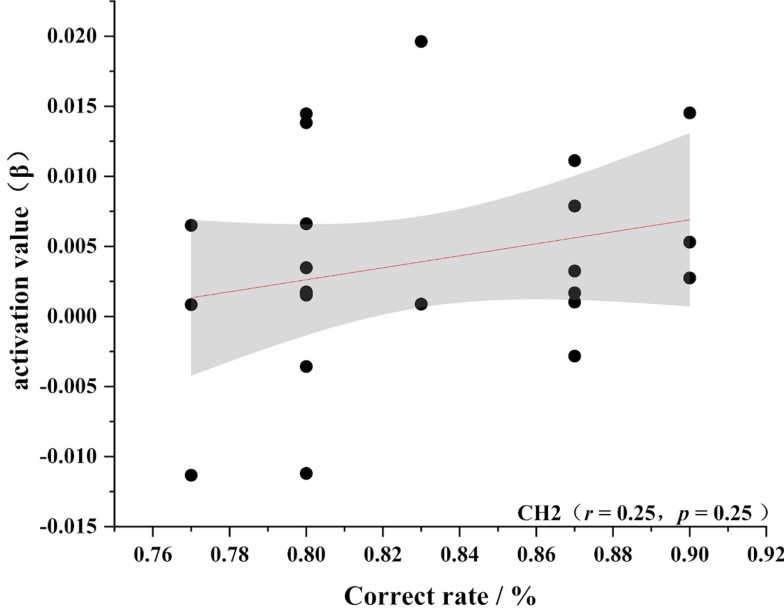

**Figure 5 Accuracy and relevance of experts group channel 2.** CH2 for channel 2.

prefrontal cortex compared to novices. These results align with prior research indicating that high visuospatial working memory capacity in experts is strongly associated with prolonged specialized training (*Lee, Ding & Chan, 2023*).

Volleyball is an open-ended sport that requires not only proficient motor skills but also the ability to respond to unpredictable environmental changes, demanding significant

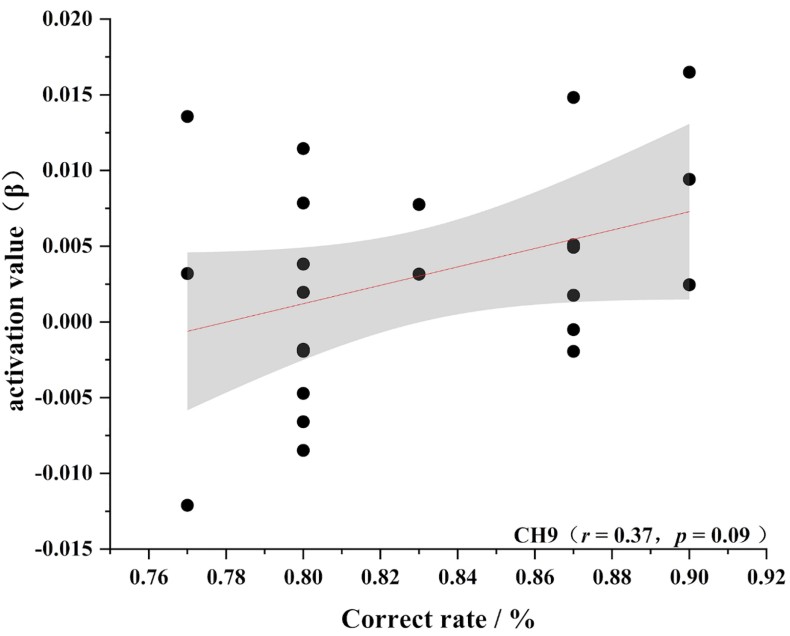

**Figure 6 Accuracy and relevance of experts group channel 9.** CH9 for channel 9.

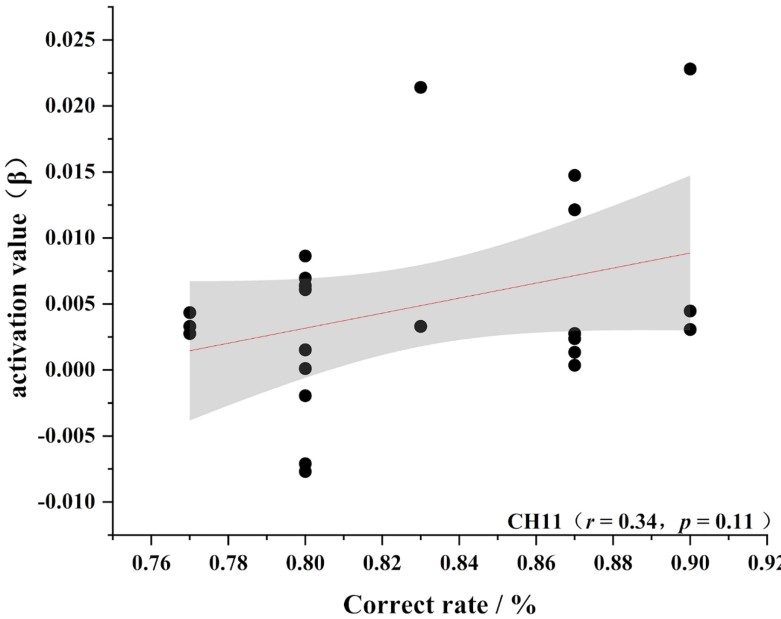

**Figure 7 Accuracy and relevance of experts group channel 11.** CH11 for channel 11.

cognitive input (*Tanida & Sakatani, 2013*) and a higher spatial memory capacity (*Figueroa-Vargas et al., 2020*). Athletes must adapt to dynamic external conditions, and specialized skills training enhances the efficiency of information processing in short-term memory, thereby improving task performance (*Jang et al., 2009*). In volleyball, players process substantial spatial information within a short time frame, requiring them to

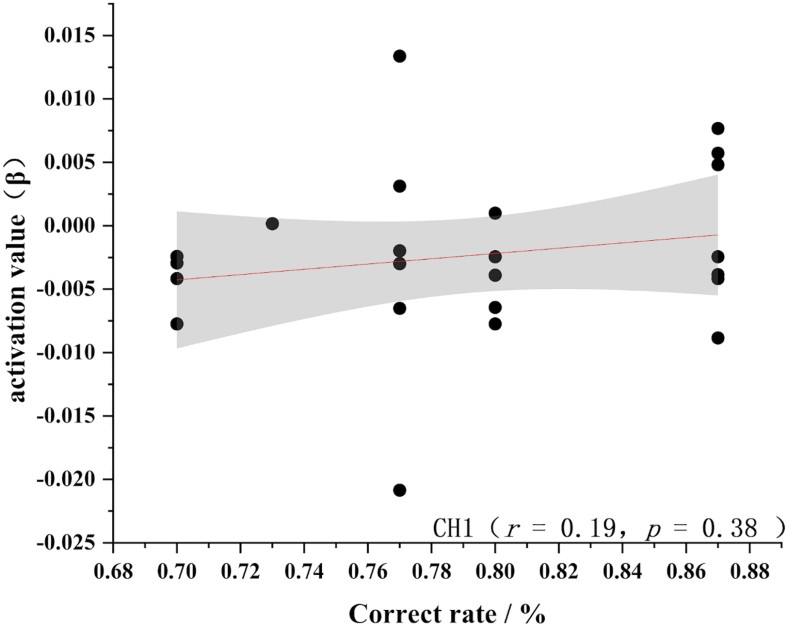

**Figure 8  Accuracy and relevance of novice group channel 1.** CH1 for channel 1.

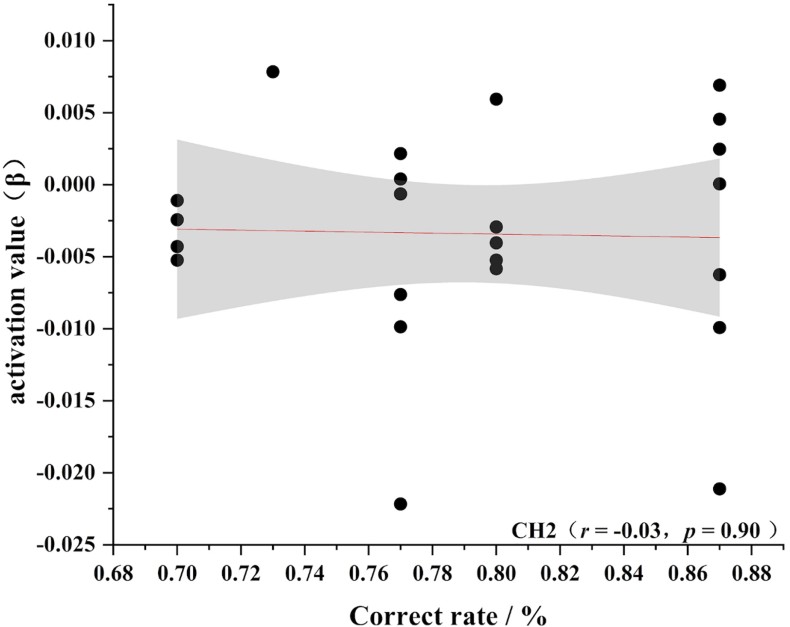

**Figure 9  Accuracy and relevance of novice group channel 2.** CH2 for channel 2.

analyze, judge, and apply this information effectively during matches. Through specialized pre-match practice, athletes store relevant information in their long-term memory, which can later be utilized during gameplay. The acquisition of domain-specific skills and knowledge enriches their experience and knowledge base. This study confirms that

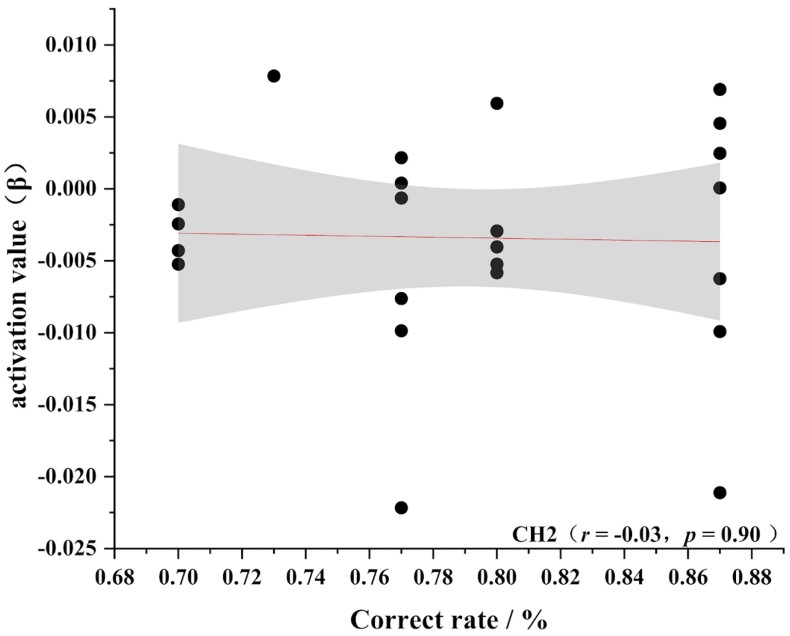

**Figure 10 Accuracy and relevance of experts group channel 9.** CH9 for channel 9.

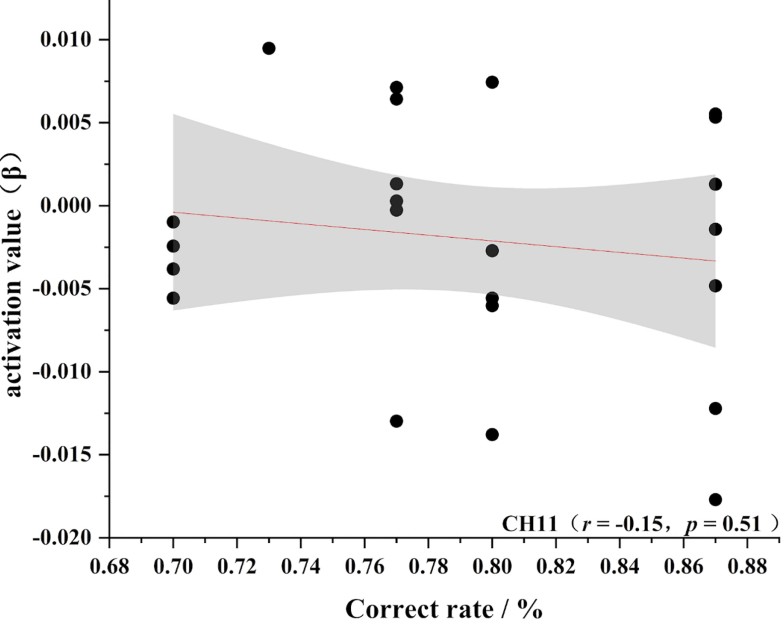

**Figure 11 Accuracy and relevance of experts group channel 11.** CH11 for channel 11.

frequent application of such knowledge facilitates faster encoding, storage, and retrieval of information from long-term memory, enabling efficient and accurate decision-making (*Lee, Ding & Chan, 2023*). This automated information processing allows experts to perform effectively in visuospatial working memory tests (*Tanida & Sakatani, 2013*).

This study also provides neurological evidence that long-term expert training induces differences in brain activation across various regions. As noted, the expert group exhibited heightened activation in specific brain areas, particularly the dorsolateral prefrontal cortex and the ventromedial prefrontal cortex, which are integral to coordinating working memory, attention, and cognitive control (*Ngetich et al., 2022*; *Sable et al., 2021*). During intense matches, specialists consistently activate task-related brain regions in a structured and focused manner, forming accurate and efficient receptor networks for motor decision-making. These networks leverage expertise stored in long-term memory and motor instructions to handle complex decisions (*Lucia, Bianco & Russo, 2023*).

In volleyball training, technical and tactical rehearsals, such as serving, blocking, and spiking, require athletes to quickly search for and recall the spatial positions of both the ball and players. Such repetitive spatial memory exercises significantly enhance visuospatial memory. Furthermore, our findings show increased activation in the right prefrontal cortex, which supports maintaining stimuli in working memory and optimally mobilizing cognitive resources (*Tanida & Sakatani, 2013*; *Seo et al., 2012*). Prolonged specialized training fosters a more coherent and efficient network of task-related brain regions, facilitating the transfer of cognitive skills acquired during training to general cognitive domains. This process not only alters the brain's cognitive processing patterns but also enhances the blood-brain barrier and neuronal activity (*Barnes & Corkery, 2018*). Consequently, expert athletes demonstrate improved visuospatial working memory, evidenced by higher levels of prefrontal cortex activation. The findings suggest that prolonged volleyball-specific training reshapes spatial cognitive processing patterns in practitioners (*Lucia, Bianco & Russo, 2023*).

We also observed an interesting trend: while the study results showed no significant differences in correlations, the behavioral indicator (correctness) exhibited a stronger correlation with prefrontal activation in the expert group compared to the novice group. Based on this observation, we hypothesize that motor training enhances the dynamics of cortical activation. In general cognitive tasks, athletes often outperform non-athlete (*Zhang et al., 2024*) due to improved working memory (*Wang et al., 2023*), attention allocation (*Gao & Zhang, 2023*), and decision-making (*Zhao et al., 2022*). However, there is no clear linear relationship between brain activation patterns and these behavioral performances (*Yu et al., 2022*). This finding suggests that the superior performance of athletes in general cognitive tasks may not solely stem from direct enhancements in brain activity but is more intricately associated with years of accumulated motor experience (*García et al., 2024*; *Yarrow, Brown & Krakauer, 2009*).

Although our findings do not directly establish causality, they provide additional evidence supporting the relationship between improved motor performance and changes in brain activity over time. A limitation of this study is the general cognitive task employed, which does not allow for a simultaneous comparison between the characteristics of the motor-scene visuospatial working memory task and the general visuospatial working memory test. This aspect could be explored in greater detail in future research.

## CONCLUSION

Overall, volleyball practice was effective in improving visuospatial working memory capacity. The main performance was attributed to the fact that participants who practiced volleyball for a long period of time had faster operational processing speeds while showing higher accuracy in visuospatial working memory tests. In addition, we hypothesized that long-term volleyball practice may have promoted plasticity changes in the brain, with dorsolateral prefrontal and frontal pole regions exhibiting optimized spontaneous neural activity during visuospatial working memory tasks. These findings provide new perspectives and mechanistic explanations for how long-term exercise training enhances cognitive performance, revealing the potential value of volleyball practice in cognitive function enhancement.

## ACKNOWLEDGEMENTS

We would like to express our gratitude to the volleyball players who participated in this study. The authors thank students for the given aid in data capture.

### Funding

The authors received no funding for this work.

### Competing Interests

The authors declare that they have no competing interests.

### Author Contributions

- Wen Zhang conceived and designed the experiments, performed the experiments, analyzed the data, prepared figures and/or tables, and approved the final draft.
- Jingru Liu conceived and designed the experiments, analyzed the data, prepared figures and/or tables, authored or reviewed drafts of the article, and approved the final draft.
- Fangfang Hu conceived and designed the experiments, analyzed the data, authored or reviewed drafts of the article, and approved the final draft.
- Yang Liu conceived and designed the experiments, performed the experiments, analyzed the data, prepared figures and/or tables, and approved the final draft.
- Chao Kan performed the experiments, analyzed the data, prepared figures and/or tables, authored or reviewed drafts of the article, and approved the final draft.

### Human Ethics

The following information was supplied relating to ethical approvals (*i.e.*, approving body and any reference numbers):

The Shaanxi Normal University granted Ethical approval to carry out the study within its facilities (Approval number: SNNU2023301).

## Data Availability

The raw measurements are available in the Supplemental File.

## Supplemental Information

Supplemental information for this article can be found online at http://dx.doi.org/10.7717/peerj.19153#supplemental-information.

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
