# Peer review of "Modulatory mechanisms of long-term volleyball practice on visuospatial working memory capacity: an fNIRS study"

_PeerJ, doi:10.7717/peerj.19153_

## Round 0.1 · original submission · Major Revisions

Dear author,

Please ensure that all of the following information is complete and accurate before resubmitting. In addition, I suggest you to do the reviewer letter.

Reviewer 1 ·

Basic reporting

The manuscript has a good structure and provides organized raw data; however, it could benefit from proofreading.
I notice that the media 'PeerJ_Human_Subject_Permission_Letter,' does not have the subject's signature. Please correct this.

Experimental design

The design is appropriate for the aims of the study, but some points need clarification

Validity of the findings

More work needs to be done, mainly in the results. In my view, the authors made statements in the conclusion that cannot be supported by the results.

Additional comments

Introduction
In lines 61-64, you provided the aims of your study. However, you repeated the same aim with different words in lines 126-129. Consider consolidating the aims into the final paragraphs of the introduction.
In line 66, consider using a term more relevant to practitioners instead of "specialist."

Methods:
In line 140, specify the number of experts and novices. I only discerned that your sample had 40 players because of Table 1.
In line 172, Table 1 does not contain information about ROI.
In lines 172-175, only six ROIs were described (although eight were cited).
The legend of Figure 2 does not explain the relevance of the red and blue colors.
Consider providing confidence measures, such as ICC, for fNIRS and cognitive testing.

Results
In lines 206-208, specify the compared groups: ...showed that the correct rate was significantly lower in the novice group than in the expert group.
In line 224, you mentioned three channels, but four were described.
In lines 228-231, you stated that variables were correlated with each other, although without statistical significance. Given that you chose frequentist statistics to report the results, only discuss positive or negative correlations when the statistics are significant.
Table 1 does not display information about novice sport level and average years of training. Provide a reason for this omission.
Table 2 should show the actual p-value and Cohen’s d for each comparison.
Figure 3 should display the actual p-value, Cohen’s d, and the variation measure used (e.g., standard deviation, standard error). Explain in the title or legend what "R" and "L" means.
Consider enlarging Figure 4 by displaying two images per line.
Why did you use only four channels? Other channels exhibited statistical differences and different distributions, which were not shown. If there is a reason for this, please explain.

Discussion
The discussion primarily focuses on a theoretical perspective. More information could be provided about the results and other studies in the literature.

Conclusion
I'm not convinced that the statement in lines 314 and 315 was supported by the results. While the roles of the regions correlated and exhibited differences between groups, caution is warranted due to the study's experimental design. Additionally, Pearson’s correlation did not show statistical significance.

Reviewer 2 ·

Basic reporting

.

Experimental design

.

Validity of the findings

.

Additional comments

I am grateful for the opportunity to evaluate the manuscript about volleyball, experience, and brain activation. After reading the manuscript, it was observed that:

1 – In the abstract, the authors use the methods section to state that they were convincing. When discussing the methods, authors must be specific and indicate the age of the participants and the measurements taken, among other variables that are important for the research. Avoid stating anything about the results.

2 – The authors do not indicate the statistical values that must be presented in the abstract.

3 – In the introduction, several statements do not indicate which author is responsible for them. Example: “The temporary storage and processing of spatial information by individuals is a fundamental issue in cognitive psychology research. Visuospatial working memory is a crucial aspect of spatial cognitive ability. It is responsible for generating, manipulating, and storing visual images to locate objects in the environment and understand spatial relationships between objects and the environment.” and “This information is then stored in short-term visuospatial working memory and matched with existing long-term memory to make accurate decisions. Therefore, this study aims to explore and analyze the visuospatial working memory characteristics of volleyball players to determine whether sports experience improves their visuospatial working memory capacity.” and “Research in the field of sport indicates that deliberate practice specialists develop specialized visuospatial working memory characteristics.” Among other statements.

4 – At the end of the introduction, it is unclear which knowledge gap will be filled. As the hypothesis suggests, it seems that this will be another study with novice and experienced athletes, as there are already several studies in this area.

5 – In the methods, the definition of experience does not converge with what is pointed out in the literature. Participating in the first league does not give the athlete expertise. Furthermore, the average age is 21, which may suggest that these athletes are not experienced. Authors must indicate the athletes' experience in years. Therefore, the shortest (novice) and longest (expert) experience periods and the quality of this experience must be shown.

6 – This is not a flaw in the research, but the authors only assessed cognition in a general test. Here, I leave the suggestion of cognitive analysis in a modality-specific test.

7 – I suggest that writing should be made easier in the results by indicating which top result to show with which group.

8 – In the discussion, there are many statements without indicating the reference author. This happened in the introduction. Furthermore, the authors state that the novices processed irrelevant signals, which cannot be confirmed from the data collection. To verify that irrelevant signals were processed, the authors should use eye tracking. Another point of attention is that the authors discuss the results from research that analyzed working memory in tasks that simulate sports demands, which is not desirable to the research design. The authors should discuss the results in light of other research that analyzed cognition in generalist tests.

Given the above, I suggest that the manuscript be rejected.

Reviewer.

Reviewer 3 ·

Basic reporting

The number or participants in inconsistent, 46 in the abstract and 40 in the rest of the article. Please, clarify that number of participants.

Line 75-76, “Conversely, studies (…)” please, cite the referred studies in the end of the sentence, so it became more clear.

Line 96-99, the sentence “Research shows (…)” is too long and contains important information to understand the investigation. I suggest splitting into two sentences to clarify the idea of not having a linear relation between the expertise and cortex activation

The keywords have different sizes and types of letters.

The last paragraph of the introduction (line 126) starts “based on these questions”. What questions do you refer to?

Please, justify the text on the Methods, point “participants” (lines 147-152); “cognitive testing” (lines 154-165) and results sections (211-225).

Experimental design

Suggestion: In methods section use the following paper to explain the level of expertise of different participants: Swann, C., Moran, A., & Piggott, D. (2015). Defining elite athletes: Issues in the study of expert performance in sport psychology. Psychology of sport and exercise, 16, 3-14.

On table 1 “Basic information table of the subjects”, please clarify the information about the weekly training frequency (hours/day) in the expert group. Do they really practice 10.2h per day?´

Please add in the tittle the information (M ± SD) as it is in the table 2.

Validity of the findings

The objective of the study is: “(…) investigates the influence of visuospatial working capacity on spatial perception and utilization in athletes during competition”. The main findings underling “the cognitive advantages and neural distinction of high-level volleyball players and hints at the transformative potential of extended training (…). So, the main finding does not match with the main objective of the study. Probably it is better to re-structure the objective of the study.´

The raw data file in .xls runs correctly and it is clear.

---

## Round 0.2 · Minor Revisions

Dear Author's

We would like to express our gratitude for submitting your manuscript entitled "Modulatory Mechanisms of Long-Term Volleyball Practice on Visuospatial Working Memory Capacity: An fNIRS Study" to our journal.

The reviewer acknowledged the quality of your work and identified a few areas that could be enhanced to further strengthen your argument and contribution to the scientific community. Their suggestions and critiques were valuable, and thus, we kindly request that you revise your manuscript according to the reviewers' suggestions.

Should you have any questions or require further clarification regarding the revisions requested, please do not hesitate to contact us. We once again appreciate your work and collaboration in this process.

Best regards,

Alexandre Medeiros

Reviewer 1 ·

Basic reporting

The article needs corrections in some structures.

Experimental design

no comment

Validity of the findings

Statistical reports can be improved.

Additional comments

I appreciate your effort to address my suggestions. However, some points still need attention.

Figure 2 does not explain itself. Please insert the meaning of the blue and red colors in the figure’s legend.

When I asked for ICC for fNIRS and cognitive testing, I was referring to a priori validation of these methods. I assume there are validity/reliability papers for these methods.

In your response letter, you wrote: “An insignificant result does not mean that there is no relationship between the two; it is just that our current data and analytical methods did not find a clear link between them. This may be because our sample size is not large enough or there are some limitations in the experimental design.” I think there is some confusion here. Indeed, a non-significant result does not mean that the correlation does not exist; it could be due to power issues. However, it also does not mean that the effect exists. To address your two hypothetical situations: (a) if your sample size is not large enough (and thus you do not have enough power), you cannot make any judgment about "non-significant" results. (b) If there are significant limitations in your experiment that could affect your results, you need to align your expectations/results/conclusions with these limitations and avoid making statements stronger than what your study allows. Finally, if you strongly believe in your hypotheses and need to include that in your analyses, you should consider using Bayesian statistics instead of frequentist. Please provide references if you have reasons to support the correlation without significant results using a frequentist approach.

---

## Round 0.3 · Major Revisions

Dear Author,

Please revise the manuscript considering the input of reviewer 1.

Thank you.

Best regards.

Reviewer 1 ·

Basic reporting

The font of the letters in lines 133-136 is different compared to the rest of the text.

Experimental design

no comment

Validity of the findings

no comments

Additional comments

I believe that performing ICC in your study may not be necessary. My request is for a study that validates your method of analysis. Since you are using fNIRS, I assume that fNIRS has validated outputs, meaning it measures with accuracy and precision. Without prior validation studies or a strong theoretical explanation, we cannot be certain about the reliability of the results.

Regarding your second question, I believe that removing your results simply because they were not significant would be a mistake. All your statistical analyses and hypotheses should be defined a priori. Frequentist statistics require this because you cannot accept a result solely based on its significance; there must be a theoretically grounded explanation. This is a crucial step in your research. Testing a hypothesis multiple times can sometimes yield a statistically significant result, but if you accept that without a prior hypothesis, you are essentially formulating hypotheses after seeing the data, which is not acceptable in frequentist theory.

A non-significant statistical result can occur for many reasons, but removing it from your research simply because you are unable to explain it is, in my view, incorrect. If you defined your hypotheses before performing the statistical test, you need to report the results, even if they are non-significant.

---

## Round 0.4 · Minor Revisions

Dear Authors,

Please revise the manuscript considering the final reviewer suggestions.

Thank you.

Best regards.

Reviewer 1 ·

Basic reporting

no comment

Experimental design

no comment

Validity of the findings

no comment

Additional comments

no comment

Reviewer 3 ·

Basic reporting

The introduction could be improved by shortening the paragraphs and eliminating redundancies. Additionally, attention should be paid to refining the grammar throughout the introduction. The author should also focus on better text organization and ensure that the entire text is properly justified.

Experimental design

Please include the number of novice participants in the methodology section; I believe there are 23, but this should be clearly stated in the text. The main goal of the study is to compare experienced and inexperienced volleyball players, with the authors concluding that volleyball practice influences visual memory. To be more precise, wouldn't it be important to create a new category of "beginner volleyball players" or consider the athletic background of the novice group? Otherwise, the observed differences could be attributed to general physical activity, rather than volleyball specifically.

Validity of the findings

The discussion section also needs improvement. The authors frequently revert to a general literature review, resembling the introduction. This section should be more concise and focused on the main ideas, directly comparing the key findings with previous studies.

---

## Round 0.5 · Minor Revisions

Dear Authors,

Please revise the manuscript considering the suggestions by reviewer 3.

Thank you.

Best regards.

Reviewer 3 ·

Basic reporting

The document has been revised to improve the overall quality of the English language. Grammar, vocabulary, and sentence structure have been enhanced to ensure clarity, professionalism, and readability. This refinement makes the content more polished and effective in conveying its message.

Experimental design

The methodology has been clarified, making it easier to understand, and all previous errors have been corrected. These updates ensure the content is both accurate and well-structured.

Validity of the findings

However, the discussion section remains unclear, with repeated sentences and an inconsistent sequence of information. Improvements are needed to help the reader distinguish between the presentation of specific investigation results and the authors' efforts to establish links with previous studies. This restructuring would enhance clarity and logical flow.

---

## Round 0.6 · accepted · Accept

Dear Authors,

Congratulations on the work performed in this manuscript. The decision is to accept, although, some improvements (for example format details) are necessary in the proof phase.

Thank you.

Best regards.